# Left Bronchial Isomerism with Right-Sided Tracheal Bronchus: A Rare Case Report

**DOI:** 10.3390/diagnostics13040751

**Published:** 2023-02-16

**Authors:** Kyungsoo Bae, Kyung Nyeo Jeon

**Affiliations:** 1Department of Radiology, Institute of Health Sciences, Gyeongsang National University School of Medicine, Jinju 52727, Republic of Korea; 2Department of Radiology, Gyeongsang National University Changwon Hospital, Changwon 51472, Republic of Korea

**Keywords:** trachea, bronchi, tracheal bronchus, bronchial isomerism, computed tomography

## Abstract

The tracheal bronchus is a congenital bronchial branching anomaly defined as an aberrant bronchus arising in either the trachea or a main bronchus. Left bronchial isomerism is characterized by two bilobed lungs, bilateral long main bronchi, and both pulmonary arteries passing superiorly to their respective upper lobe bronchi. Left bronchial isomerism with a right-sided tracheal bronchus is a very rare combination of tracheobronchial anomalies. It has not been previously reported. We present multi-detector CT findings of a left bronchial isomerism with a right-sided tracheal bronchus of a 74 year old man.

Congenital tracheobronchial branching anomalies are known to be present in approximately 1% of the general population [1]. These anomalies are often overlooked. However, they need to be identified prior to surgery or intubation to avoid complications. We report imaging findings of left bronchial isomerism combined with a tracheal bronchus.

A 74 year old male patient was hospitalized with a progressive weakness of his arms that began seven months prior to hospitalization. Based on neurologic studies, he was diagnosed with amyotrophic lateral sclerosis. As he also complained of exertional dyspnea upon admission, an evaluation was performed for the presence of thoracic lesions. A chest X-ray showed unremarkable findings. A contrast-enhanced chest CT scan was also performed, showing no parenchymal infiltrates or pulmonary thromboembolism. However, there were tracheobronchial branching anomalies. At the level of the carina, an aberrant bronchus was noted, arising from the right main bronchus (tracheal bronchus) (Figure 1a). A coronal, reformatted image showed that both main bronchi were long, with right main bronchus resembling the left bronchus morphologically (Figure 1b). A reconstructed, 3D CT image of the tracheobronchial tree revealed a symmetrical bronchial branching pattern of both bronchi. Similar to the left upper lobe bronchus, the right middle lobe bronchus (B4 and B5) arose with the upper lobe bronchus (Figure 1c). A tracheal bronchus arising from the right main bronchus corresponded to a displaced apical segmental bronchus (B1) of the right upper lobe. A superimposed, three dimensional (3D) CT image of the tracheobronchial tree and pulmonary arteries showed that both upper lobe bronchi were hyparterial. The upper lobe bronchus arose below the point where the ipsilateral pulmonary artery crossed the main bronchus (Figure 1d). Both lungs were bilobed (Figure 1e). There was no minor fissure in the right lung. Further systemic evaluation did not reveal any congenital cardiovascular disease or abdominal visceral malposition. Therefore, the patient was diagnosed as having an isolated bronchial branching anomaly, left bronchial isomerism with a right-sided, eparterial tracheal bronchus. Despite bronchial anomaly, it was determined that the patient’s respiratory symptoms were attributed to progressive amyotrophic lateral sclerosis.

To the best of our knowledge, this is the first report of a patient presenting left bronchial isomerism and tracheal bronchus. Tracheal bronchus is a collection of bronchial variations arising from anywhere from either the trachea or main bronchus. Most tracheal bronchi arise within 2 cm of the carina [2]. They are directed to the upper lobe territory. Tracheal bronchus might be rudimentary, displaced, or supernumerary [3]. Embryologically, there are three major hypotheses explaining anomalous bronchi: reduction, migration, and selection theories. Rudimentary tracheal bronchus can be explained by the reduction theory, whereas displaced or supernumerary tracheal bronchus can be explained by the migration or selection theories [2,3]. In the present case, the displaced tracheal bronchus ventilated the right upper lobe apical segment (B1). Visceral heterotaxy refers to abnormally arranged internal organs, which are mirror images that present bilaterally in an otherwise asymmetric anatomy. The patients can be classified into either right or left isomerism depending on the morphology of the paired structures. Bronchial situs can be determined from the relationship between the upper lobe bronchus and the pulmonary artery. The left upper lobe bronchus arises below the point where the left pulmonary artery crosses the left main bronchus (hyparterial), whereas the right upper lobe bronchus arises above the point where the right pulmonary artery crosses the right main bronchus (eparterial) [3]. The present case demonstrated bilateral long main bronchi, both hyparterial upper lobe bronchi, and bilobed lungs, consistent with left bronchial isomerism. Commonly, there is a concordance between bronchial, atrial, and abdominal visceral arrangement in patients with situs anomalies [4]. However, our case showed isolated left bronchial isomerism without atrial isomerism, congenital heart disease, or visceral heterotaxy. Very few cases of complicated bronchial branching anomalies, such as combined bilateral tracheal bronchi and right bronchial isomerism, have been reported in the literature [5,6].

Most bronchial branching anomalies are asymptomatic and incidentally discovered, particularly in adults. However, knowledge and understanding of bronchial abnormalities have important implications when performing therapeutic interventions or surgeries [7]. Multi-detector CT is the gold-standard tool for detecting congenital tracheobronchial anomalies with high spatial resolution, allowing for the visualization of the pathologies on multiple planes using various reformation techniques. A recent study regarding the anatomical variants identified on a large number of CT scans of COVID-19 patients has identified tracheal bronchus with a prevalence of 0.2% [8]. In those patients, special attention should be paid to avoid endotracheal tubes obstructing the tracheal bronchus when intubation or mechanical ventilation is needed.

In conclusion, we report a detailed imaging analysis of a rare case with left bronchial isomerism and right-sided tracheal bronchus. Physicians who perform bronchoscopy or bronchopulmonary surgeries should identify bronchial variations and their relationships with pulmonary vessels on preoperative CT using multiplanar reformation and 3D reconstruction.

## Figures and Tables

**Figure 1 diagnostics-13-00751-f001:**
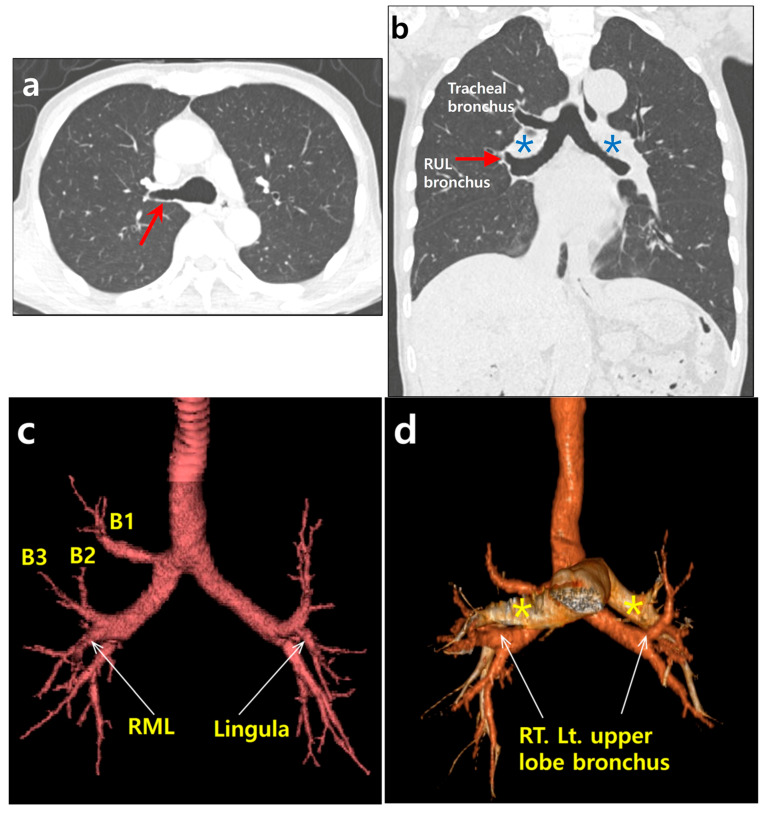
A 74 year old man with tracheal bronchus and left bronchial isomerism. (**a**) An axial chest CT image with lung window setting showing an aberrant bronchus (tracheal bronchus, arrow) arising from the right main bronchus at the level of carina. (**b**) A coronal, reformatted chest CT image with lung window setting showing symmetrical long both main bronchi resembling the left bronchus morphologically. Right upper lobe bronchus (RUL) is located below the point where the pulmonary artery (asterisk) crosses the main bronchus as on the left (hyparterial). Note tracheal bronchus directing towards the right upper lobe. (**c**) Reconstructed, 3D CT image of the tracheobronchial tree revealing both upper lobe bronchi in the same position. Similar to lingular division (arrow) of the left upper lobe bronchus, the right middle lobe bronchus (RML, arrow) arises with the right upper lobe bronchus. Displaced apical segmental bronchus (B1, thin arrow) arises from tracheal bronchus. Posterior and anterior segmental bronchi (B2 and B3) arise from right upper lobe bronchus. (**d**) Superimposed 3D CT image of the tracheobronchial tree and pulmonary arteries (asterisks) showing that both upper lobe bronchi (arrows) are hyparterial. (**e**) Sagittal reformatted CT images with lung window setting showing bilobed left and right lungs (arrows indicate both major fissures). There is no minor fissure in the right lung.

## Data Availability

Data are contained within the article. No new data were created or analyzed in this study.

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
