# Peer review of "Left Bronchial Isomerism with Right-Sided Tracheal Bronchus: A Rare Case Report"

_diagnostics, 2023, doi:10.3390/diagnostics13040751_

Round 1

Reviewer 1 Report

The  paper was like Case report. There is no generalization of the paper. Methodology is poor

1.      What is the main question addressed by the research?

The main objective of this work is to identify the isomerism in the Tracheal bronchus through CT Images.
2. Do you consider the topic original or relevant in the field, and if
so, why? Yes, But the article was a case report.
3. What does it add to the subject area compared with other published
material? NO

4. What specific improvements could the authors consider regarding the
methodology?

 Rare case analysis

5. Are the conclusions consistent with the evidence and arguments presented and do they address the main question posed?

No the conclusion needs improvement.
6. Are the references appropriate? No

7. Please include any additional comments on the tables and figures.

  The entire paper has to rewritten for better understanding.

Author Response

 Answers to Reviewers’ Comments and Suggestions

 (Reviewer 1)

The  paper was like Case report. There is no generalization of the paper. Methodology is poor

1. What is the main question addressed by the research?

The main objective of this work is to identify the isomerism in the Tracheal bronchus through CT Images.

A) We thank the reviewer for pointing this out and we agree with the reviewer. Therefore, we have added the main objective of this work as suggested by the reviewer.

2. Do you consider the topic original or relevant in the field, and if

so, why? Yes, But the article was a case report.

A) We thank the reviewer for pointing this out. We have revised the topic of our manuscript.

3. What does it add to the subject area compared with other published

material? NO

A) We thank the reviewer for pointing this out. We have added more references and compared our case with other published materials.

4. What specific improvements could the authors consider regarding the

methodology? Rare case analysis

A) We thank the reviewer for pointing this out and we agree with the reviewer. Therefore, we have revised our manuscript as suggested by the reviewer.

5. Are the conclusions consistent with the evidence and arguments presented and do they address the main question posed? No the conclusion needs improvement.

A) We thank the reviewer for pointing this out. We have revised our Conclusion.

6. Are the references appropriate? No

A) We thank the reviewer for pointing this out. We have added more references.

7. Please include any additional comments on the tables and figures.The entire paper has to rewritten for better understanding.

A) We thank the reviewer for pointing this out. We have replaced the original Figure 1e with a new figure having better quality. We also added more details. In addition, we have asked a Professional English Editing Service (Harrisco) to improve the language for better understanding.

We thank the reviewers for their valuable comments and suggestions. These comments and suggestions have improved the quality of our manuscript significantly.

Reviewer 2 Report

None.

Author Response

(Reviewer 2)

Comments and Suggestions for Authors

None.

A) We thank the reviewer for reviewing our manuscript. We have revised our manuscript, references, and figures.

Reviewer 3 Report

A case of tracheal bronchus was reported. This is an extremely rare variation worthy of publication. However, there are a few comments to improve the paper. One major concern is that key literature was not reviewed. 

Average prevalence of tracheal bronchus was not reported. Please add the overall prevalence of tracheal bronchus to the manuscript (https://www.sciencedirect.com/science/article/abs/pii/S0003497520318543)

Embryological basis of the tracheal bronchus has to be added briefly (https://pubmed.ncbi.nlm.nih.gov/11158647/)

A recent paper identified two cases of tracheal bronchus in 2/999 (0.2%) COVID-19 patients (https://pubmed.ncbi.nlm.nih.gov/35385153/). In one of the two cases, tracheal bronchus was accompanied by the sternal foramen, another variation of the sternum. This is another case of combined anomaly that should be added.

Author Response

(Reviewer 3)

1. A case of tracheal bronchus was reported. This is an extremely rare variation worthy of publication. However, there are a few comments to improve the paper. One major concern is that key literature was not reviewed. 

Average prevalence of tracheal bronchus was not reported. Please add the overall prevalence of tracheal bronchus to the manuscript (https://www.sciencedirect.com/science/article/abs/pii/S0003497520318543)

A) We thank the reviewer for pointing this out and we agree with the reviewer. Therefore, we have added the reference suggested by the reviewer.

2. Embryological basis of the tracheal bronchus has to be added briefly (https://pubmed.ncbi.nlm.nih.gov/11158647/)

A) We thank the reviewer for pointing this out and we agree with the reviewer. Therefore, we have added the embryological basis for tracheal bronchial anomalies and cited the reference suggested by the reviewer.

3. A recent paper identified two cases of tracheal bronchus in 2/999 (0.2%) COVID-19 patients (https://pubmed.ncbi.nlm.nih.gov/35385153/). In one of the two cases, tracheal bronchus was accompanied by the sternal foramen, another variation of the sternum. This is another case of combined anomaly that should be added.

A) We thank the reviewer for pointing this out. We have added and cited the reference suggested by the reviewer.

We thank the reviewers for their valuable comments and suggestions. These comments and suggestions have improved the quality of our manuscript significantly.

Round 2

Reviewer 1 Report

Some of the corrections are carried out by the authors. The section 3 and design methods are to be included.

Author Response

(Answers to reviewer’s comments)

As we have already mentioned the paper is like a case study of a patient and hence, the author has to generalize the finding through scientific methods of statistical tests or clinical finding apart from investigation procedure as given by CT images.

A) To help readers understand, we have added a general description on our patient's anomalies.

To attain a generalized diagnostic decision process the author has to explain the standard experiment procedure for many patients rather than a single case or a group of cases.

A) We have added an explanation of the gold standard diagnostic method for congenital tracheobronchial anomalies.

We thank the reviewer for his/her valuable comments and suggestions. These comments and suggestions have improved the quality of our manuscript.